# Bacterial Tomato Pathogen *Ralstonia solanacearum* Invasion Modulates Rhizosphere Compounds and Facilitates the Cascade Effect of Fungal Pathogen *Fusarium solani*

**DOI:** 10.3390/microorganisms8060806

**Published:** 2020-05-27

**Authors:** Lv Su, Lifan Zhang, Duoqian Nie, Eiko E. Kuramae, Biao Shen, Qirong Shen

**Affiliations:** 1Jiangsu Provincial Key Lab of Solid Organic Waste Utilization, Jiangsu Collaborative Innovation Center of Solid Organic Wastes, Educational Ministry Engineering Center of Resource-Saving Fertilizers, Nanjing Agricultural University, Nanjing 210095, China; sulv000@126.com (L.S.); zhanglf1397@163.com (L.Z.); 2016103120@njau.edu.cn (D.N.); shenqirong@njau.edu.cn (Q.S.); 2Microbial Ecology Department, Netherlands Institute of Ecology (NIOO-KNAW), Droevendaalsesteeg 10, 6708 PB Wageningen, The Netherlands; E.Kuramae@nioo.knaw.nl; 3Ecology and Biodiversity, Institute of Environmental Biology, Utrecht University, Padualaan 8, 3584 CH Utrecht, The Netherlands

**Keywords:** *Ralstonia solanacearum* invasion, phenolic acids, *Fusarium solani*, co-occurrence

## Abstract

Soil-borne pathogen invasions can significantly change the microbial communities of the host rhizosphere. However, whether bacterial *Ralstonia solanacearum* pathogen invasion influences the abundance of fungal pathogens remains unclear. In this study, we combined high-throughput sequencing, qPCR, liquid chromatography and soil culture experiments to analyze the rhizosphere fungal composition, co-occurrence of fungal communities, copy numbers of functional genes, contents of phenolic acids and their associations in healthy and bacterial wilt-diseased tomato plants. We found that *R. solanacearum* invasion increased the abundance of the soil-borne pathogen *Fusarium solani*. The concentrations of three phenolic acids in the rhizosphere soil of bacterial wilt-diseased tomato plants were significantly higher than those in the rhizosphere soil of healthy tomato plants. In addition, the increased concentrations of phenolic acids significantly stimulated *F. solani* growth in the soil. Furthermore, a simple fungal network with fewer links, nodes and hubs (highly connected nodes) was found in the diseased tomato plant rhizosphere. These results indicate that once the symptom of bacterial wilt disease is observed in tomato, the roots of the wilt-diseased tomato plants need to be removed in a timely manner to prevent the enrichment of other fungal soil-borne pathogens. These findings provide some ecological clues for the mixed co-occurrence of bacterial wilt disease and other fungal soil-borne diseases.

## 1. Introduction

Bacterial wilt caused by *Ralstonia solanacearum* is a destructive and widespread soil-borne disease [1,2], and it is not effectively controlled in the field [3]. The first visible symptoms of bacterial wilt are usually wilting of the youngest leaves. A sticky, milky-white exudate is observed on the surface of freshly-cut sections from infected stems, and it represents dense masses of bacterial cells in the stem. Moreover, it is known that *R. solanacearum* invasion significantly affects the diversity and composition of the bacterial community in the tomato rhizosphere [4,5]. For example, *R. solanacearum* disturbs tomato rhizosphere bacterial communities, leading to reductions in bacterial diversity and the abundance of nonpathogenic bacteria [4]. The composition of the bacterial community in the tomato rhizosphere was also changed via the enrichment of Alphaproteobacteria and depletion of Gammaproteobacteria under *R. solanacearum* invasion [5]. In addition, foot rot of tomato caused by the fungal pathogen *Fusarium solani* is another soil-borne disease that limits tomato production [6]. *F. solani* can infect tomato roots and young foliage. A dark brown lesion of about 1 inch long can be visible on the tap root or a main lateral root, and the infected foliage shows interveinal chlorosis and necrotic spotting [7]. However, whether *R. solanacearum* invasion can affect fungal communities, specifically whether changes in fungal pathogens occur in the rhizosphere of tomato plants, remains unknown.

Phenolic acids from root residues and root exudates play important roles in the soil microbial community assembly [8,9]. For example, exogenous salicylic acid (0.5 mM) was added every three days to *Arabidopsis thaliana* leaves until an inflorescent meristem formed, and the enrichment of *Flavobacterium* spp. and *Terracoccus* spp. and depletion of *Mitsuaria* spp. were found in the endophytic bacterial community of the roots in the treated plant samples [9]. Moreover, phenolic acids can affect soil-borne pathogen growth [10,11]; for example, *R. solanacearum* can utilize salicylic acid (inducement of host defense response) to stimulate its growth and enhance its virulence in tomato plants [11]. Ferulic acid is a novel inducer of the type 3 secretion system (T3SS) in *R. solanacearum*, which can promote the *R. solanacearum* infection process in tomato plants [12]. Similarly, tomato root exudates stimulate the microconidia germination of both *Fusarium oxysporum* f. sp. *lycopersici* and *Fusarium oxysporum* f. sp. *radicis-lycopersici* fungal pathogens [13]. Furthermore, the colony forming units (CFUs) of the *F. oxysporum* population were increased in soil amended with *p*-coumaric acid at concentrations of 0.1, 0.25, 0.5 and 1.0 µmol/g soil without plants [14]. Vanillic acid, *p*-hydroxybenzoic acid and coumalic acid at the concentration of 6.5 mg/L could facilitate the infection of peanut seeds by *F. solani* in vitro [15]. The above information suggests that phenolic acids may play a role in stimulating soil-borne pathogen growth.

Microbial networks are used to indicate positive and negative associations among microorganisms [16] and microbial community complexity [17]. For example, *R. solanacearum* reduces the complexity of bacterial networks in the rhizosphere of tomato plants with bacterial wilt disease [4]. Similarly, *R. solanacearum* invasion reduces the average degree and link number in the bacterial community of tobacco plants [18]. However, whether *R. solanacearum* invasion can affect fungal network co-occurrences within the tomato plant rhizosphere has not been investigated. Here, we determined the effects of *R. solanacearum* invasion on the abundance of the fungal tomato plant pathogen *F. solani* via qPCR. The effects of *R. solanacearum* invasion on the composition and networks of the fungal community were determined by high-throughput sequencing. Our aim is to determine whether the fungal community composition, specifically the fungal pathogens, and the fungal networks are affected by *R. solanacearum* invasion in the rhizosphere of tomato plants. We hypothesized that *R. solanacearum* invasion may change the rhizosphere environment, thereby leading to changes in the fungal community in the tomato rhizosphere.

## 2. Material and Methods

### 2.1. Experimental Field Site and Sampling Regime

The field experiment was conducted in Qilin Town (118°57′ E, 32°03′ N), a vegetable production center in the urban area of Nanjing, China. The soils were Luvisols (FAO). The experimental field had been continuously planted with tomatoes for more than 5 years before the experiment, and tomato bacterial wilt disease occurred naturally. The experimental field was divided into 30 plots. To weaken the previous differences in the fungal community of the field soils, we randomly and widely took 72 tomato plants from 24 plots (12 plots for healthy plants and 12 plots for diseased plants). The detailed sampling method is listed as follows: At the end of the harvest (approximately 14 weeks for tomato plants, 11 October 2016), twelve plots with healthy or bacterial wilt-diseased tomato plants (level-four disease severity [19]) were selected to sample. Within each plot, rhizosphere soil samples from three tomato plants were collected and mixed into a single sample (Figure 1). After the plants were dug out, the roots were carefully shaken to remove the loosely adhering soil, and the remaining attached soil was carefully collected as rhizosphere soil using a sterile brush. In total, 24 rhizosphere soil samples were obtained for soil DNA extraction.

Another batch of healthy and bacterial wilt-diseased tomato plants were sampled in the soil, in which tomato and strawberry plants were planted under rotation conditions, and strawberries were grown in the previous planting cycle (October 2016 to March 2017). These tomato plants were sampled at the tomato florescence stage (26 April 2017) in the same vegetable production center of Qilin Town. The sampling method was similar to that described above. Briefly, seven plots with healthy or bacterial wilt-diseased tomato plants at level-four disease severity were selected and three tomato plants were mixed into a single sample in each plot. After pooling, 14 rhizosphere soil samples were obtained for soil DNA extraction to determine the copy number of *fliC* (*R. solanacearum*) and ITS I (*F. solani*).

To test that the tomato wilt symptoms were caused by *R. solanacearum*, the pathogen *R. solanacearum* was isolated from diseased tomato stems [20] using the selective media SMSA (Selective Medium South Africa). Three *R. solanacearum* isolates were randomly selected from the SMSA selective medium. Ten tomato seedlings (4-leaf stage) were inoculated with each *R. solanacearum* isolate, and all tested isolates can cause tomato bacterial wilt disease according to Koch’s Rule. DNA of the isolates from the selective media was extracted and used as the template to amplify the 16S rRNA gene by PCR. The primers and PCR conditions are listed in Appendix A respectively. The PCR products were sequenced with ABI 3730 by the QinKe Company (Nanjing, China), and the sequences were compared with the NCBI database via BLAST.

### 2.2. Determination of Phenolic Acids in the Rhizosphere Soil

Phenolic acids from the rhizosphere soil were extracted as previously described [21], with slight modifications. Briefly, 5 mL of 1 M NaOH was added to 1 g rhizosphere soil in a flask, which was shaken in the dark for 24 h (210 rpm, 30 °C). After centrifugation at 8000× *g* for 10 min, the supernatant was transferred to another flask and acidified to pH 2.5 with 5 M HCl to precipitate humic acid. After 2 h, the supernatant was centrifuged at 8000× *g* for 10 min and then lyophilized. The lyophilized powder was dissolved in 1 mL deionized water and then filtered through a 0.45 μm Millipore membrane for HPLC analysis (Agilent 1200, Santa Clara, CA, USA) as previously described [22,23] with slight modifications. The analytical conditions were as follows: chromatographic column, SB-Aq (4.6 × 250 mm); column temperature, 40 °C; mobile phase flow velocity, 1.0 mL min^−1^; detection wavelength, 280 nm; and injection volume, 20 µL. Methanol (A) and 2% (*v*/*v*) acetic acid solution (pH 2.59) (B) were used as the mobile phases with a gradient elution (B: 100% (0 min) →70% (27 min) →37% (42 min) →25% (52 min) →0% (55 min) →end (60 min)). The standard phenolic acids used for the HPLC analysis included phthalic acid, *p*-hydroxybenzoic acid, vanillic acid, ferulic acid, cinnamic acid and salicylic acid (Sigma-Aldrich Co., LLC., Burlington, MA, USA). The phenolic acids in the rhizosphere soils were identified by comparing their retention times with those of matching standards. The concentrations of each tested phenolic acid in the soil extracts were obtained based on peak areas using the external standard method and expressed as micrograms per gram of dry soil.

To test whether the increased phenolic acid levels from the rhizosphere soil of bacterial wilt-diseased tomato can facilitate the enrichment of *F. solani*, we performed a soil culture experiment. The soil culture experiment involved two treatments: soil amendments to high and low concentrations of phenolic acids based on the phenolic acid levels in diseased (D) and healthy (H) rhizosphere soils, respectively, and a control (C) with soil but no phenolic acid amendments (Figure 1). Each treatment had five replicates. The soils were collected from the field as described above before the tomatoes were planted, and then mixed, air dried and sieved (20 mesh screen). The mixed solution of three phenolic acids was added to 250 g soil from each treatment at a high concentration (45.24 μg g^−1^ for *p*-hydroxybenzoic acid, 8.27 μg g^−1^ for vanillic acid and 37.18 μg g^−1^ for ferulic acid) and a low concentration (10.95 μg g^−1^ for *p*-hydroxybenzoic acid, 4.38 μg g^−1^ for vanillic acid and 7.7 μg g^−1^ for ferulic acid). The final soil water content was maintained at 30% of its water holding capacity. Then, 250 g of soil from each treatment was divided into 5 fractions as replicates, and each replicate containing 50 g of soil was placed into a plastic bottle (100 mL volume). After incubation at 25 °C for 5 days, 0.25 g of soil from each replicate was randomly sampled for DNA extraction. The DNA extraction procedure was the same as previously described.

### 2.3. Determination of the Soil pH, Total Carbon and Nitrogen

The rhizosphere soil pH, as well as the total carbon and nitrogen were determined as previously described [24]. Briefly, the soil pH (soil/water = 1:2.5) was measured with a Leici PHS-3C basic pH meter (Shanghai, China). The total carbon (total C) and nitrogen (total N) were determined with an elemental analyzer according to the manufacturer’s instructions (Elementar vario EL III, Hanau, Germany).

### 2.4. DNA Extraction and Sequencing

Total genomic DNA was isolated from the tomato plant rhizosphere soil (0.25 g) using the PowerSoil DNA Isolation Kit (Mobio Laboratories, Carlsbad, CA, USA) following the manufacturer’s protocol. The microbial community composition was characterized via MiSeq sequencing. The PCR primers ITS5-1737-F (5-GGAAGTAAGTCGTAACAAGG-3) and ITS2-2043-R (5-GCTGCGTTCTTCATCGATGC-3) were used to amplify the ITS1 region between 18S and 5.8S rRNA. The primers used for the final sequencing of the PCR amplicon products included the appropriate Illumina adapters for each sample. The PCR products from each sample were used to construct a sequencing library with the Illumina TruSeq DNA Sample Preparation Kit (Illumina). For each sample, the ITS1 PCR amplicons were sequenced by the Personalbio Company, Shanghai, China. The raw sequences were submitted to the NCBI Sequence Read Archive (SRA) under the submission ID SUB5746768.

The sequence data were mainly processed on the USEARCH platform [25]. Briefly, sequences with a quality score lower than 0.5 or a length shorter than 200 nt and singletons were discarded. Noisy sequences were filtered, chimeras were inspected, and an operational taxonomic unit (OTU) cutoff was assigned at the 97% identity level. The OTU abundance of each sample was standardized using the lowest value (69,207) of the sequencing depth. Representative sequences of each OTU were selected and classified by the RDP classifier against the UNITE Fungal ITS database for fungi, which corresponded to 40% for fungi [26]. The other unclassed taxa were classified by the Warcup Fungal ITS database [27] and corresponded to 40%. Functional information from the annotation result of the UNITE Fungal ITS database for the OTUs was predicted by FUNGuild [28].

### 2.5. Quantitative PCR

The copy numbers of the subunit of the flagellar filament gene (*fliC*) (*R. solanacearum*), 16S rRNA gene (total bacteria), ITS I (*F. solani*), internal transcribed spacer (ITS) region (total fungi) and genes related to the production of the antagonistic substances lipopeptide (*sfr* (surfactin), *fen* (fengycin) and *itu* (iturin)) and polyketides (*dfn* (difficidin)) were quantified by quantitative PCR in all samples. Quantitative PCR (qPCR) assays were performed using the SYBR Premix Ex Taq^TM^ (Perfect Real Time) Kit (Takara Biotechnology Co., Dalian, China) with the ABI StepOne^TM^ Real-Time PCR System (Applied Biosystems, ‎Waltham, MA, USA). Each reaction was performed in a 20 µL volume. Detailed primer information and PCR steps are listed in Appendix A, respectively. Standard curves were developed by serially diluting the plasmids with known positive inserts to final concentrations of 10^2^ to 10^7^ gene copies µL^−1^. qPCR efficiencies ranged from 90% to 105%, and the *R*^2^ values were greater than 0.99. Three independent technical replicates were used for each sample.

### 2.6. Statistical Analyses

Fungal Shannon diversity was calculated from the rarefied fungal OTU table using Mothur software. Based on the Bray–Curtis index dissimilarity (relative abundance data) and Jaccard index dissimilarity (presence–absence data), a principal coordinate analysis (PCoA) of the fungal community structure was performed using the vegan package in R. To test whether the healthy and diseased tomato plant rhizosphere soil samples had different centroids, a PERMANOVA test was performed using the Adonis function in vegan. The homogeneity of the multivariate dispersions was checked with the betadisper function in vegan. The indicator fungal genus (top 20) for healthy and diseased tomato plant rhizosphere soil samples were identified with the indicspecies package in R.

Significant differences in the number of gene copies, soil factors and Shannon diversity were assessed using Student’s *t*-test, and the data conformed to a normal distribution according to the Shapiro–Wilk test in R. The data on the copy number of ITS I (*F. solani*) in the soil culture experiment were subjected to one-way ANOVA and then to Tukey’s test for multiple comparisons.

Mantel tests were used to identify correlations between the fungal compositions and soil environmental factors, the copy number of *fliC* (*R. solanacearum*) and relative abundances of OTUs belonging to the potential fungal pathogens as well as correlations between the copy numbers of antagonistic genes and copy number of *fliC* (*R. solanacearum*) and ITS I (*F. solani*) using the vegan package for R. The association between the Shannon diversity and the copy numbers of *fliC* (*R. solanacearum*) and ITS I (*F. solani*) was determined by the cor.test function in R.

We performed a random forest analysis to determine the regression associations between soil factors and dominant OTUs (top 20) (Appendix A). Random forest analysis can provide high prediction accuracy by building the decision trees based on bootstrapped samples. This analysis is a nonlinear statistical and nonparametric method without prior distributional assumptions. Certain samples used to train the model are called in-bag data, and the other samples are termed out-of-bag data. Trees fully grown are used to predict the out-of-bag data, and the importance of the variable is obtained by randomly permuting the values of that variable for the out-of-bag data and calculating increases in the mean squared error (%IncMSE). The higher the IncMSE value, the more important the variable. In our study, a random forest analysis was performed with 999 permutations using the randomforest and rfPermute packages and visualized using Gephi [29].

The OTU abundances (top 300) were used to build a microbial network by the function sparcc with 100 permutations in Mothur software. Edges whose *p* value was < 0.001 were retained. The link number of nodes (OTU degree), clustering coefficient and visualized networks were assessed using Gephi software [29]. In this study, OTUs with degrees higher than 10 were selected as hub taxa (highly connected nodes). The link number distributions of the fungal networks were calculated with the density function in R.

## 3. Results

### 3.1. Identification of the Isolates from the Stem of Bacterial Wilt-Diseased Tomato Plant

The BLAST results showed that the isolated strains from the diseased tomato stems shared 99.9% identity with the *R. solanacearum* GMI1000 strain (race 1, biovar 3). The phylogenetic tree of the isolates is shown in Appendix A. The sequences were deposited in the NCBI GenBank (submission number: SUB5692774).

### 3.2. Fungal Community Diversity, Structure and Composition in Healthy and Bacterial Wilt-Diseased Tomato Plant Rhizospheres

Because high abundances (up to approximately 73.47%) of the ITS sequences belonging to the tomato plant were observed in certain rhizosphere soil samples, these samples (three samples for healthy plants and three samples for diseased plants) were removed in the analysis of fungal communities. *R. solanacearum* invasion significantly reduced the Shannon diversity of the fungal community (*p* = 0.02) (Figure 2a). The results of the PCoA showed that a clear difference was found between the fungal community structures of bacterial wilt-diseased and healthy tomato plant rhizospheres (Figure 2b). The Adonis test also confirmed the significant effect of *R. solanacearum* invasion on the fungal community (*p* = 0.002, *R* = 0.95), although the betadisper results were not significant (*p* = 0.12, F = 2.68), suggesting that samples from healthy and diseased tomato plant rhizospheres had the same dispersions. In addition, based on the Jaccard index dissimilarity of presence–absence data, *R. solanacearum* invasion significantly changed the fungal community of the tomato plant rhizosphere (Appendix A).

At the order level, Hypocreales and Sordariomycetidae were enriched approximately 2.91-fold and 8.79-fold, while Sordariales was depleted approximately 3.71-fold in the fungal communities of bacterial wilt-diseased plant rhizospheres relative to healthy tomato plant rhizospheres (Figure 2c). An indicator species analysis was performed to test the characteristic taxa (at the genus level) in the fungal communities of bacterial wilt-diseased and healthy tomato plant rhizospheres. The indicator taxa of one treatment indicated that such taxa are characteristic in that treatment. The potential plant pathogens *Fusarium*, *Gibellulopsis*, *Nectria* and *Plectosphaerella* were the indicator taxa for bacterial wilt-diseased tomato plants (Table 1). Eight fungal genera, including beneficial fungi *Mortierella* and *Chaetomium*, were the indicator taxa for healthy tomato plants.

### 3.3. Functional Genes in the Tomato Rhizosphere

The qPCR-determined copy numbers of the 16S RNA gene (total bacteria), ITS (total fungi), *fliC* (*R. solanacearum*) and ITS I (*F. solani*) in the rhizosphere of bacterial wilt-diseased tomato plants were significantly higher than those in the rhizosphere of healthy tomato plants (Figure 3). Specifically, the copy number of *fliC* (*R. solanacearum*) and ITS I (*F. solani*) in the diseased tomato plant rhizosphere was 679-fold and 10.51-fold higher than that in the healthy tomato plant rhizosphere, respectively. However, the copy numbers of the functional genes surfactin (*sfr*), fengycin (*fen*), difficidin (*dfn*) and iturin (*itu*), which are related to the synthesis of antagonistic substances against *Fusarium* [30], were significantly higher in the healthy tomato plant rhizosphere than in the diseased tomato plant rhizosphere. In addition, the copy numbers of *fliC* (*R. solanacearum*) and ITS I (*F. solani*) were significantly and negatively correlated to the copy numbers of theses antagonistic genes (Figure 6a).

The ratio of functional genes to total bacteria and fungi showed that the relative abundances of *fliC* and ITS I in the rhizosphere soil of bacterial wilt-diseased tomato plants were also significantly higher than those in the rhizosphere soil of healthy tomato plants (Appendix A).

According to the results of another batch of healthy and bacterial wilt-diseased tomato plants in the soil with a planted-strawberry history, the copy numbers of *fliC* (*R. solanacearum*) and ITS I (*F. solani*) in the rhizosphere soil of wilt-diseased tomato plants were approximately 1817-fold and 9-fold higher than those in the rhizosphere soil of healthy tomato plants, respectively (Appendix A). These findings also showed that *R. solanacearum* invasion can increase *F. solani* abundance in the rhizosphere soil.

### 3.4. Fungal Co-Occurrence in Healthy and Bacterial Wilt-Diseased Tomato Plant Rhizospheres

Due to the significant differences in the fungal community compositions of the healthy and bacterial wilt-diseased tomato plant rhizospheres, we further analyzed the differences between the fungal co-occurrence of both treatments. The results showed clear differences in the co-occurrence structure and topology between healthy and bacterial wilt-diseased tomato plant rhizospheres. There were 242 nodes and 772 edges in the healthy fungal community co-occurrence (Figure 4). However, the diseased fungal co-occurrence contained 214 nodes and 554 edges. The clustering coefficient of the fungal co-occurrence of diseased tomato plant rhizospheres was also lower than that of the fungal co-occurrence of healthy tomato plant rhizospheres. Moreover, the number of hub (highly connected nodes) taxa (33) in the fungal co-occurrence of the healthy tomato plant rhizosphere (Appendix A) was higher than that in the fungal co-occurrence of the wilt-diseased tomato plant rhizosphere, which contained 10 hub taxa (Appendix A). The potential plant growth-promoting fungi (PGPF) hub taxa OTU146 belonging to *Mortierella* were found only in the fungal co-occurrence of the healthy tomato plant rhizosphere. Although certain OTUs belonging to the beneficial fungus *Chaetomium* were found in both hub OTUs of the fungal co-occurrence of healthy and diseased tomato plant rhizospheres, the negative link between the dominant OTU5 (approximately 12.67%) belonging to the beneficial fungus *Chaetomium* and OTU7 belonging to the potential pathogen fungus *Gibellulopsis* was only found in the fungal co-occurrence of the healthy tomato plant rhizosphere (Appendix A).

The node degree distribution indicated that there were higher link numbers among nodes in the fungal co-occurrence of the healthy tomato plant rhizosphere than in that of the diseased tomato plant rhizosphere (Appendix A). The node link numbers of orders shared by the healthy and bacterial wilt-diseased tomato plant rhizosphere fungal networks were compared (Appendix A). The results showed that the order Onygenales had the most link numbers in the fungal networks of both healthy and wilt-diseased tomato plant rhizospheres. The link numbers of Microascales and Branch06 in the fungal network were slightly more in the diseased samples than in the healthy samples. The link numbers of other order nodes in the fungal network of the healthy tomato plant rhizosphere were higher than those in the fungal network of the bacterial wilt-diseased tomato plant rhizosphere. Overall, the fungal network of the healthy tomato plant rhizosphere was more complex and contained more hub taxa than that of the bacterial wilt-diseased tomato plant rhizosphere.

Because certain OTUs existed in the bacterial community of only diseased or healthy samples, to eliminate the effect of the number of OTUs on fungal network complexity, the fungal networks of the healthy and diseased rhizospheres were constructed with shared OTUs. The results showed that the numbers of network nodes and edges and the clustering coefficient of the healthy plant network were also higher than those of the diseased-plant network (Appendix A).

### 3.5. Changes in Soil Factors in Healthy and Bacterial Wilt-Diseased Tomato Plant Rhizospheres

Three phenolic acids (*p*-hydroxybenzoic acid, vanillic acid and ferulic acid) were detected in healthy and wilt-diseased tomato plant rhizospheres (Appendix A). Their contents in the wilt-diseased tomato plant rhizosphere were significantly higher by approximately 1.88-fold to 4.82-fold than those in the healthy tomato plant rhizosphere (Table 2). Specifically, the content of ferulic acid in the wilt-diseased tomato plant rhizosphere was 4.8-fold higher than that in the healthy tomato plant rhizosphere. In addition, the pH levels and total C and N were significantly higher in the wilt-diseased tomato plant rhizosphere than in the healthy tomato plant rhizosphere. Mantel tests showed that all tested soil factors had significant correlations with the fungal community composition (Appendix A).

### 3.6. Identification of Soil Factors Determining Dominant OTUs

The results of the random forest analysis showed that OTU16, belonging to the potential soil pathogen *F. solani*, was affected positively by vanillic acid and total C (*p* = 0.04, *R^2^* = 0.65) (Figure 5 and Appendix A). Regardless of the taxa cutoff (taxonomic assignment), OTU3 and OTU7, belonging to the potential plant pathogens *Plectosphaerella cucumerina* and *Verticillium dahliae*, were significantly and positively affected by the total N (*p* = 0.01, *R^2^* = 0.23) and *p*-hydroxybenzoic acid (*p* = 0.04, *R^2^* = 0.19), respectively.

According to the results of the Mantel tests, the increased abundances of *R. solanacearum* (*fliC*), fungal OTUs belonging to the potential fungal pathogens (*Fusarium*, *Plectosphaerella* and *Verticillium*) had significant associations with the fungal community structure (Figure 6b). Moreover, the copy number of *fliC* (*R. solanacearum*) and ITS I (*F. solani*) was negatively correlated to the fungal diversity (Shannon diversity index; Figure 6a). Overall, the increased contents of phenolic acid, total N and total C in the wilt-diseased rhizosphere soil may facilitate the growth of potential fungal pathogens, which in turn results in changes in the fungal communities.

Because the potential fungal pathogen *F. solani* had the highest taxa cutoff, a soil culture experiment was performed via qPCR to confirm whether the phenolic acid contents from the wilt-diseased tomato rhizosphere soils can stimulate *F. solani* growth. The results showed that the maximum copy number of ITS I (*F. solani*) was found under the phenolic acid concentration of the rhizosphere soil from bacterial wilt-diseased tomato plants (Appendix A), thus confirming that the concentrations of phenolic acids in the wilt-diseased rhizosphere can stimulate the growth of the soil pathogen *F. solani*.

## 4. Discussion

Tomato bacterial wilt reduces yields and leads to unbalanced rhizosphere microbial communities. Previous studies focused on the negative effects of *R. solanacearum* invasion on the bacterial communities [4,5]; however, whether *R. solanacearum* invasion influences the fungal community of the tomato plant rhizosphere soil is still unknown. Here, we showed that *R. solanacearum* invasion greatly affected the fungal community composition by increasing the abundance of the soil pathogenic fungus *F. solani* and decreasing the fungal diversity and copy numbers of the functional genes related to antagonistic substances and complexity of fungal co-occurrence in the tomato plant rhizosphere soil.

In our study, *R. solanacearum* invasion facilitated the enrichment of *F. solani*, a pathogenic fungus, in the rhizosphere of tomato plants (Figure 3). It is likely that *R. solanacearum* can cause brownish discoloration in the root vascular system, thereby leading to varying degrees of root decay and resulting in the release of phenolic acids. Indeed, we found that the three tested phenolic acids were increased in the rhizosphere of diseased tomato plant (Table 2) and the indigenous fungal pathogen *F. solani* was significantly enriched in soil amended with such levels of phenolic acids (Appendix A). Previous studies showed that certain phenolic acids have a positive effect on fungal pathogen growth at suitable concentrations [22,23]. In addition, the environment of rotten roots may be greatly different from that of heathy roots in the rhizosphere soil, such as by releasing the highly toxic substance putrescine. It is known that *R. solanacearum* is strongly competitive and demonstrates phenotypic changes in response to different environments [31] as well as having a high putrescine tolerance [32]. Thus, by competing with other microbes, *R. solanacearum* can reduce the soil microbial diversity during invasion [4]. The decrease in microbial diversity may provide more niches for microbes that are suitable for the rotten root environment, such as *F. solani* [33]. Moreover, the copy number of the functional genes (*sfr*, *fen*, *dfn* and *itu*) related to the synthesis of antagonistic substances against *F. solani* [30] was significantly decreased in the wilt-diseased rhizosphere soils (Figure 3) and had negative associations with the copy number of ITS I (*F. solani*) (Figure 6a), suggesting that the decreased abundance of antagonistic substances produced by such genes stimulated the growth of *F. solani*.

To test the robustness of our results, we sampled bacterial-wilt diseased and healthy samples in another soil. The results confirmed that *R. solanacearum* invasion can increase the abundance of *F. solani*, regardless of the history of soil use and tomato plant growth stage, thus indicating the generalizability of our results (Appendix A). Moreover, regardless of the cutoff of annotation taxa, the relative abundances of other tomato potential fungal pathogens, such as *Plectosphaerella cucumerina* and *Verticillium dahliae*, had positive associations with the increased contents of total N and *p*-hydroxybenzoic acid (Figure 5). The findings suggest that the observed enrichment of *F. solani* may not be a special case. It is known that rotten plant tissue can stimulate soil pathogen growth [34]. Thus, a consortium of fungal pathogens was likely enriched in the rotten rhizosphere environment caused by *R. solanacearum* invasion. This result indicated an increased potential risk of host infection by fungal pathogens in the wilt-diseased soil. These potential pathogens must be quantified by qPCR in a future study.

*R. solanacearum* invasion significantly changed the fungal community structure. It is likely that the rhizosphere environment was changed by *R. solanacearum* invasion, which might have led to root decay and consequently the release of nutrients into the soil as observed in Table 2. In line with our studies, several studies have shown that phenolic acids could affect fungal communities [8,35]. In addition, *R. solanacearum* invasion increased the caffeic acid content of tomato root exudates that affect rhizosphere microbial communities [36]. Thus, the changes in nutrients, such as phenolic acids, may facilitate the enrichment of potential fungal pathogens, thereby resulting in the changes in fungal communities as observed by the associations in our model (Figure 6b). Although fake associations between enrichment of certain microbes and changes in the fungal community structure may still exist in our model (Figure 6b), and given this problem cannot be solved via the current statistical approach, our main conclusion that *R. solanacearum* invasion increases the phenolic acid contents, thereby favoring colonization by the pathogenic fungi *F. solani*, was not affected by the observed problem.

In this study, a higher soil pH was found in the rhizosphere soil of wilt-diseased tomato plants (Table 2). It is likely that the content of ammonia might have increased during the root decay process, leading to an increase in the soil pH [37]. However, no significant differences in soil pH were found between the healthy and wilt-diseased tomato plant rhizospheres in the study of Wei et al. [4]. It is likely that the sampling regime in the study of Wei et al. was different from that in our study. Tomato plant rhizosphere samples were collected every week for 12 weeks in the study of Wei et al. At the beginning of the bacterial wilt disease, the tomato plant roots were slightly rotten. However, we took samples at the end of the tomato harvest (14 weeks), and the roots of certain samples were gravely rotten. In addition, the high pH and total C content reportedly increased fungal diversity [38,39]. However, low fungal diversity was found in the wilt-diseased rhizosphere soils with a high pH and total C content. It is likely that the negative effects on fungal diversity of substances released from rotten roots may be stronger than the positive effects of high pH and total C content. In line with our study, fungal diversity was significantly reduced in the soil amended with straw at days 4 and 14 [40].

Network inference tools have been widely used to determine associations among microorganisms and the complexity of co-occurrence. The approach of this study showed that the fungal co-occurrence network of the healthy tomato plant rhizosphere soil was more complex than that of the bacterial wilt-diseased tomato plant rhizosphere soil (Figure 4). In agreement with our results, studies have shown that *R. solanacearum* invasion reduces the bacterial network complexity of tomato [4] and tobacco [18] plant rhizosphere soils. It is known that complex co-occurrences are more robust against environmental factors than simple co-occurrences [41]. Moreover, hub taxa (highly connected nodes) may play important roles in maintaining microbial community stability and coordinating many relationships throughout the microbiome [42,43,44,45]. Surprisingly, in our study, the hub taxa OTU146 belonging to *Mortierella* were found only in the rhizosphere fungal co-occurrence of healthy tomato plants. *Mortierella hygrophila* can induce vine plant defense responses to powdery mildew disease by the production of polyunsaturated fatty acids [46]. *Mortierella* has also been found in the rhizosphere soil of lettuce cultivated in soil amended with chitin [47] and in forest ecosystems [48]. These results suggest that the potentially beneficial fungi *Mortierella* may form cooperative associations with other taxa to stimulate plant host growth. For example, the combined application of arbuscular mycorrhizal fungi and *Mortierella* yielded a better effect on the increase in the shoot and root growth of *Kosteletzkya virginica* plants than the application of arbuscular mycorrhizal fungi alone [49]. The biocontrol agent *Chaetomium* has been used to inhibit plant pathogens, such as *Phytophthora palmivora* of pepper, *F. oxysporum* of tomato and *Sclerotium rolfsii* of corn [50]. In our study, *Chaetomium* was significantly and greatly enriched in the rhizosphere of healthy tomato plants (Table 1), and a negative link between *Chaetomium* and *Gibellulopsis* was only found in the rhizosphere of healthy tomato plants (Appendix A). This finding suggests that *Chaetomium* may play important roles in protecting hosts from soil-borne pathogen invasion. In addition, a few links between *F. solani* and other OTUs were found in the rhizosphere of wilt-diseased and healthy tomato plants. It is likely that the inferred co-occurrence was built by the relative abundances of OTUs, which resulted in certain biases.

*R. solanacearum* invasion reduced the fungal co-occurrence complexity (Figure 4). It is possible that the enriched microbes, such as *R. solanacearum* and *F. solani*, occupy more niches, thereby leading to the disappearance of certain members of the fungal community. A negative association between the fungal diversity and copy numbers of *fliC* (*R. solanacearum*) and ITS I (*F. solani*) was observed in Figure 6a. Moreover, low-diversity microbial communities resulted in a simple microbial co-occurrence [51]. Overall, the enriched microbes, stimulated by the nutrients from the rotten environment, such as phenolic acids, occupied more niches and resulted in a low fungal diversity and simple fungal co-occurrence (Figure 6a). However, several studies showed that the application of phenolic acids can stimulate the growth of *Fusarium* but cannot affect fungal diversity [14,52]. It is likely that other substances from rotten root environments, such as cellulose, and the decreased abundance of antagonistic genes resulted in great enrichment of *Fusarium* [53], which occupied the niches and in so doing reduced the fungal diversity. When the fungal networks were constructed with shared OTUs, the fungal network complexity of the diseased rhizosphere community was also lower than that of the healthy rhizosphere fungal network community (Appendix A). It is likely that the lower nutrient contents, such as total C and N, and the phenolic acid contents (Table 2) resulted in the enhancement of cooperative associations among microbes under resource-limited conditions in healthy tomato plant rhizosphere soils. In line with the results of our study, the application of organic amendments containing lower immediate nutrient resources resulted in more links among the fungal communities in bulk soils [17].

## 5. Conclusions

In this study, we combined high-throughput sequencing, qPCR and soil culture experiments to reveal that *R. solanacearum* invasion increased the rhizosphere phenolic acid contents and decreased the copy numbers of antagonistic genes, which led to the enrichment of a second pathogen, *F. solani*. Based on this finding, we propose suggestions for the management of plant diseases in agricultural systems. Once the symptoms of bacterial wilt disease of tomato is observed, the roots of wilt-diseased tomato plants need to be removed in a timely manner. In addition, the cascade changes in rhizosphere compounds and the abundance of potential fungal pathogens triggered by *R. solanacearum* invasion may be the main reasons for the changes in fungal communities. This finding may provide some ecological clues for the mixed occurrence of bacterial wilt disease and other fungal soil-borne diseases.

## Figures and Tables

**Figure 1 microorganisms-08-00806-f001:**
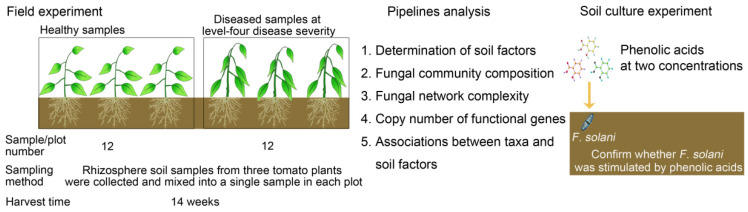
Flow diagram of the experimental design, sampling regime and pipeline analysis.

**Figure 2 microorganisms-08-00806-f002:**
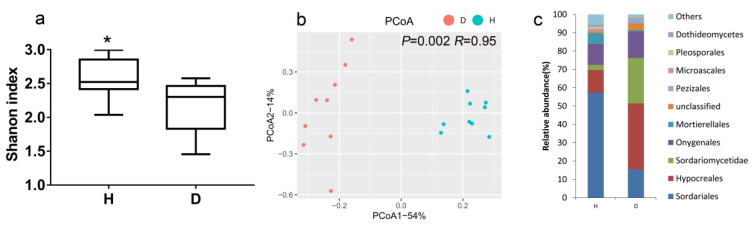
Fungal community diversity, structure and composition in healthy (H) and bacterial wilt-diseased (D) tomato plant rhizospheres. (**a**) Fungal community Shannon diversity of rhizosphere soils between healthy (H) and bacterial wilt-diseased (D) tomato plant rhizospheres. Statistical significance was determined based on Student’s *t*-test. * *p* < 0.05. (**b**) Principal coordinate analysis (PCoA) based on Bray–Curtis index dissimilarity of relative abundance data in healthy (H) and bacterial wilt-diseased (D) tomato plant rhizosphere fungal communities. (**c**) Relative abundance (%) of the orders in the rhizospheres of healthy (H) and bacterial wilt-diseased (D) tomato plants.

**Figure 3 microorganisms-08-00806-f003:**
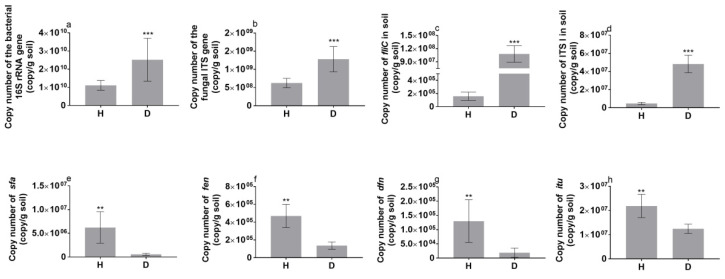
Copy number of functional genes in the rhizospheres of healthy (H) and bacterial wilt-diseased (D) tomato plants. (**a**) 16S rRNA, (**b**) ITS, (**c**) *fliC*, (**d**) ITS I, (**e**) *sfr*, (**f**) *fen*, (**g**) *dfn* and (**h**) *itu* represent total bacteria, total fungi, *R. solanacearum*, *F. solani*, surfactin, fengycin, difficidin and iturin, respectively. Statistical significance was determined based on Student’s *t*-test. *** *p* < 0.001, ** *p* < 0.01, * *p* < 0.05.

**Figure 4 microorganisms-08-00806-f004:**
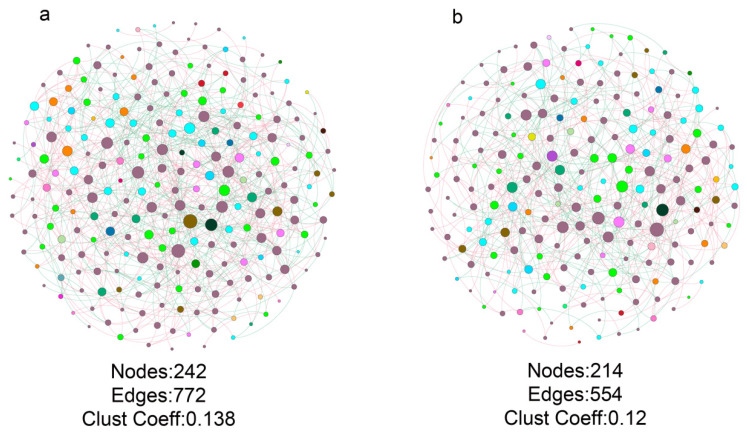
Fungal networks of healthy (**a**) and bacterial wilt-diseased (**b**) tomato plant rhizospheres. The number of nodes and edges and the clustering coefficients are shown below the networks. The node sizes represent the link numbers. Pink and blue lines represent negative and positive associations, respectively.

**Figure 5 microorganisms-08-00806-f005:**
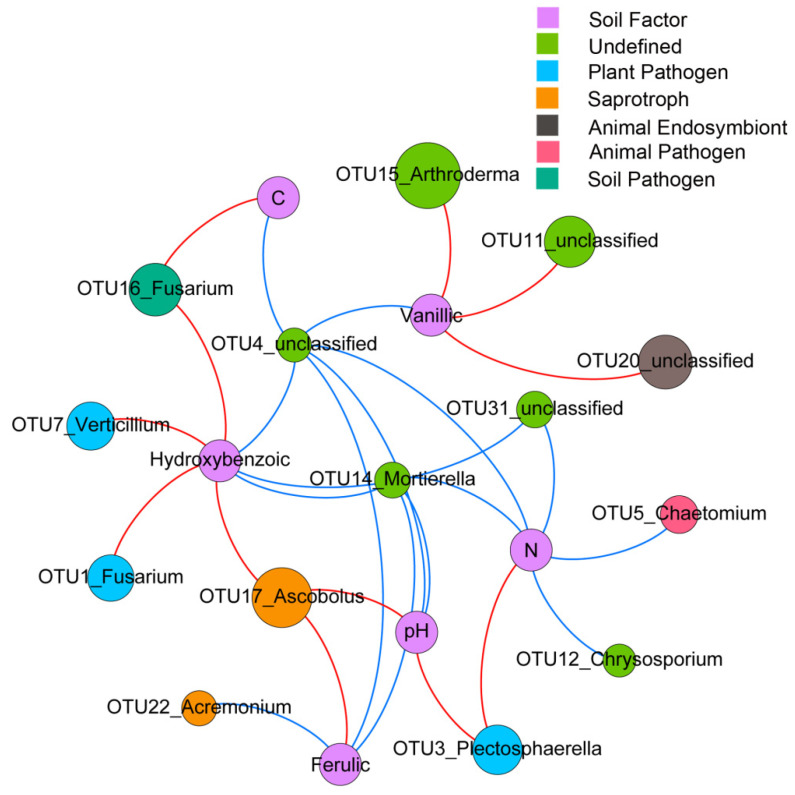
Network showing the soil factors in determining the abundance of dominant OTUs (top 20) by random forest analysis. The size of the circles represents the ratio of OTU relative abundance between diseased and healthy samples. The color of the circles represents the functional information predicted by FUNGuild. The green and red links represent the cases where soil factors negatively or positively affected the OTUs, respectively. Ferulic, vanillic and hydroxybenzoic represent ferulic acid, vanillic acid and *p*-hydroxybenzoic acid, respectively.

**Figure 6 microorganisms-08-00806-f006:**
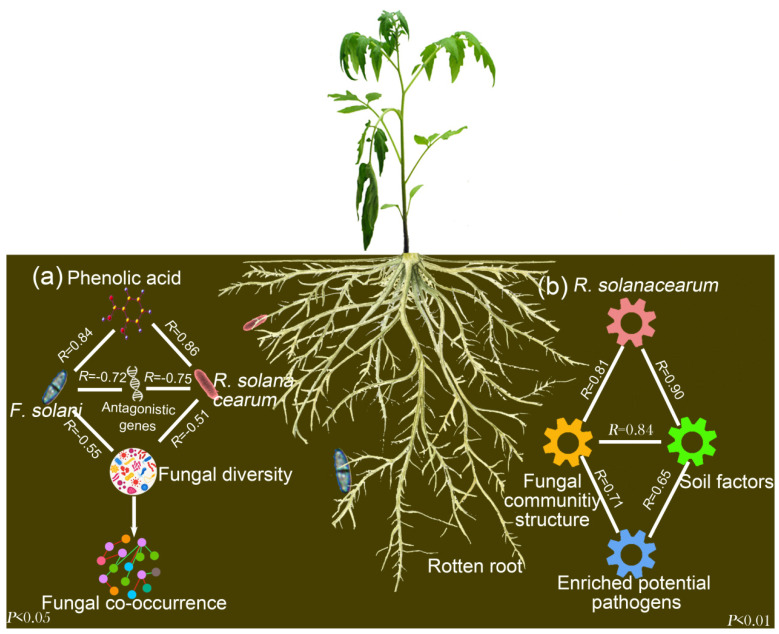
Proposed model for (**a**) *R. solanacearum* invasion, release of phenolic acids from rotten roots, decrease in antagonistic gene copy numbers, increase in *F. solani* abundance, decrease in fungal diversity and complexity of fungal co-occurrence; and (**b**) reasons for the changes in fungal community structure.

**Table 1 microorganisms-08-00806-t001:** Genus-level indicator taxa of the fungal communities in healthy (H) and bacterial wilt-diseased (D) tomato plant rhizospheres.

Genus	Relative Abundance (%)		*p*
	H	B	
*Acremonium*	1.04 ± 0.74	0.15 ± 0.22	0.001
***Arthroderma***	3.1 ± 2.29	10.46 ± 9.97	0.041
*Chaetomium*	13.29 ± 10.32	4.34 ± 3.05	0.019
*Chrysosporium*	7.08 ± 3.5	0.52 ± 0.47	0.001
***Fusarium***	32.71 ± 12.81	8.93 ± 4.11	0.001
***Gibellulopsis***	0.58 ± 0.57	3.4 ± 2.31	0.004
*Humicola*	4.29 ± 3.94	0.29 ± 0.33	0.001
*Mortierella*	5.86 ± 6.73	0.73 ± 0.73	0.002
***Nectria***	0.2 ± 0.2	2.82 ± 3.18	0.002
*Phaeoacremonium*	1.02 ± 0.81	0.12 ± 0.17	0.002
***Plectosphaerella***	2.82 ± 4.04	24.82 ± 17.91	0.003
*Pseudogymnoascus*	1.32 ± 0.66	0.15 ± 0.12	0.001
*Sarocladium*	1.33 ± 1.39	0.18 ± 0.11	0.008
*Thielavia*	32.09 ± 8.37	3.12 ± 1.72	0.001

Indicator taxa for diseased samples are marked in bold type.

**Table 2 microorganisms-08-00806-t002:** Environmental factors of healthy (H) and wilt-diseased (D) tomato plant rhizosphere samples.

Sample Group	*p*-hydroxybenzoic Acid (μg g^−1^)	Vanillic Acid (μg g^−1^)	Ferulic Acid (μg g^−1^)	pH	C (g kg^−1^)	N (g kg^−1^)
H	10.95 ± 2.13	4.38 ± 1.16	7.7 ± 0.99	6.66 ± 0.22	11.86 ± 2.85	1.69 ± 0.15
D	45.24 ± 3.93 ***	8.27 ± 1.95 ***	37.18 ± 4.83 ***	7.47 ± 0.21 ***	21.5 ± 3.76 ***	2.57 ± 0.26 ***

Statistical significance was determined based on Student’s *t*-test. *** *p* < 0.001. C and N represent the total carbon and nitrogen, respectively.

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
