# Peer review of "Bacterial Tomato Pathogen Ralstonia solanacearum Invasion Modulates Rhizosphere Compounds and Facilitates the Cascade Effect of Fungal Pathogen Fusarium solani"

_microorganisms, 2020, doi:10.3390/microorganisms8060806_

Round 1
Reviewer 1 Report
Please find my comments in the PDF file

Author Response
Answer to the reviewers:
Dear Reviewers,
Thanks for reviewing our manuscript during the epidemic period.
Major modifications were listed as follows.
- We checked the annotation information of OTUs and found large tomato ITS sequences (up to approximately 73.47%) existed in certain samples. Amplification of so many non target sequences may consume a lot of PCR materials leading to the low sequences for target sequences that may not meet the fungal community analysis condition. Thus, we deleted these samples (3 samples for healthy plants and 3 samples for diseased plants) and redo the fungal community analysis with only fungal sequences.
- Because the website of network tool Ecological Network Analyses Pipeline (MENA) cannot be opened, we used Sparcc to perform the network analysis.
Although these changes are in the new version, our main conclusion is the same.
- The new manuscript was checked by professional English editing service company (AJE)
Thanks again for your suggestions!
Abstract
- Line 25
co-occurrence of what
Answer: It means co-occurrence of fungal communities. Corrected lines30
- Line 31-34: Furthermore, the fungal community of the healthy tomato plant rhizosphere harbored a more complex fungal network with more links, nodes and hub (highly connected nodes) numbers than the fungal community of the bacterial wilt-diseased tomato plant rhizosphere.
This is too long and unclear please reformulate
Answer: We have shortened and reformulated these sentences (line36). Furthermore, a simple fungal network with fewer links, nodes and hubs (highly connected nodes) was found in the diseased tomato plant rhizosphere
- Line 35-37: These results indicate some suggestions for the management of plant diseases in agricultural systems and provide some ecological clues for the mixed co-occurrence of bacterial wilt disease and other fungal soil-borne diseases.
This is general sentence please be more specific
Answer: We have made them more specific (line38).
These results indicate that once the symptom of bacterial wilt disease is observed in tomato, the roots of the wilt-diseased tomato plants need to be removed in a timely manner to prevent the enrichment of other fungal soil-borne pathogens.
Keywords:
- Line 39: Facilitation, Fungal community, Functional genes, Phenolic acids, Co-occurrence
These keywords do not refer directly to your work, change them to increase the visibility of your work
Answer: Changed to:
Ralstonia solanacearum invasion, Phenolic acids, Fusarium solani, Co-occurrence (line44).
Introduction
- General comments to the introduction
1.1 please use the same tense in the introduction
Answer: Corrected to the same tense is used in the introduction
1.2 introduce Fusarium solani (there is no information regarding this pathogen in your introduction)
Answer: We have added the part of pathogen Fusarium solani. It contains the symptoms of foot rot of tomato caused by F. solani (line 58-62) and positive effects of phenolic acids on the infection of host by F.solani. (line 81-83)
1.3. Describe the symptoms resulted from each pathogen
Answer: The symptoms of two diseases (bacterial wilt and foot rot) were added in the introduction. line(48-51)
1.4. Do not use too long sentences
Answer: We have checked our introductions to avoid using too long sentences.
- Line 48-49: In addition, R. solanacearum invasion decreased the relative abundance of Gammaproteobacteria in the rhizosphere of healthy tomato plants.
reformulate in order to be compatible with the previous sentence
Answer: These sentences were reformulated to : “
The composition of the bacterial community in the tomato rhizosphere was also changed via the enrichment of Alphaproteobacteria and depletion of Gammaproteobacteria under R. solanacearum invasion.”
- Line 50-52: However, there are no studies on whether R. solanacearum invasion can affect the fungal communities, specifically the fungal pathogens in the rhizosphere of tomato plants.
improve the language
Answer: Changed to: “However, whether R. solanacearum invasion can affect fungal communities, specifically whether changes in fungal pathogens occur in the rhizosphere of tomato plants, remains unknown.”
- Line 54-55: For example, soil amended with salicylic acid enhanced Flavobacterium sp. and Terracoccus sp. and depleted Mitsuaria sp
please improve the sentence is this soil with or without plants? which plants? please be specific
Answer: For clarification we added (line67-71):” For example, exogenous salicylic acid (0.5 mM) was added every 3 days to Arabidopsis thaliana leaves until an inflorescent meristem formed, and the enrichment of Flavobacterium sp. and Terracoccus sp. and depletion Mitsuaria sp. were found in the endophytic bacterial community of the roots in the treated plant samples.”
- Line 56: phenolic acids can affect soil-borne pathogen growth
this is similar to the first sentence of the paragraph
Answer: The reference that supports phenolic acids, such as salicylic acid, can stimulate the grwoth of Ralstonia solanacearum, was added at the end of the sentence.
- Line 56-58: for example, tomato plants accumulate salicylic acid for protection in the presence of R. solanacearum, but this pathogen can degrade salicylic acid, enhancing its virulence in tomato plants
improve the language
Answer: The sentence was changed to: “R. solanacearum can utilize salicylic acid (inducement of host defense response) to stimulate its growth and enhance its virulence in tomato plants” (lines72-74)
- Line 63: Furthermore, the amount of F. oxysporum increases in soil amended with p-coumaric acid at concentrations of 0.1, 0.25, 0.5, and 1.0 µmol/g soil
Expressed as what? fungal DNA, fungal biomass, chitin?
again what is the plant in this soil?
Answer: The amount of F. oxysporum means the number of colony forming unit (CFU) and the soil was not planted soil. Please see lines78-81 for clarity.
“Furthermore, the colony forming units (CFUs) of F. oxysporum population were increased in soil amended with p-coumaric acid at concentrations of 0.1, 0.25, 0.5, and 1.0 µmol/g soil without plants.”
- Line 68-70: For example, R. solanacearum reduces the complexity of rhizosphere bacterial networks in tomato plants with bacterial wilt disease
You have already mentioned this in the lines 46-48 besides improve the language
Answer: In line 46-48 (old version), we have changed the sentence and highlighted the effects of R. solanacearum on the diversity and composition of bacterial community in tomato rhizosphere. Line (52) new version.
Here, we highlighted the effects of R. solanacearum on the bacterial network, and the language was improved. Please see following.
“For example, R. solanacearum reduces the complexity of bacterial networks in the
rhizosphere of tomato plants with bacterial wilt disease.” (line86-88)
- Line 70-72: Similarly, simple microbial networks with decreased average degrees and numbers of links were found in a bacterial wilt-diseased tobacco plant rhizosphere
this is not clear
Answer: We have made it clear. Please see following. (Line 88)
Similarly, R. solanacearum invasion reduces the average degree and link number in the bacterial community of tobacco plants
- Line 72-74: However, whether R. solanacearum invasion can affect fungal network co-occurrences of the tomato plant rhizosphere has not been investigated.
You have already mentioned this in the lines 50-52
Answer: In lines 50-52 (old version), we highlighted the effect of R. solanacearum invasion on the composition of fungal community. Here, we highlighted the effect of R. solanacearum invasion on fungal network. Please see following(line90-91).
However, whether R. solanacearum invasion can affect fungal network co-occurrences within the tomato plant rhizosphere has not been investigated.
- Line 74-78: Here, we determined the effects of R. solanacearum invasion on the abundance of the fungal tomato plant pathogen Fusarium solani and on the rhizosphere fungal community composition and fungal co-occurrence in bacterial wilt-diseased and healthy tomato plant rhizospheres under field conditions by using qPCR and high-throughput sequencing, respectively.
This too long, please use short sentences and improve the language
Answer: We have shortened and improved the language. Please see following (line91-95).
Here, we determined the effects of R. solanacearum invasion on the abundance of the fungal tomato plant pathogen F. solani via qPCR. The effects of R. solanacearum invasion on the composition and networks of fungal community were determined by high-throughput sequencing.
- Line 78-84: Phenolic acid concentrations of the rhizosphere soil of bacterial wilt-diseased plants were determined, and a soil culture experiment was performed to test whether such phenolic acid concentrations simulated the growth of a second pathogen, F. solani. To enhance the result robustness, the copy numbers of fliC (R. solanacearum) and ITS I (F. solani) of tomato plant rhizosphere were determined in another soil, in which strawberry had been grown in last planting cycle.
this is materials and methods
Answer: These sentences were deleted in induction.
- Line 84-86: We hypothesized that R. solanacearum invasion stimulates a second fungal pathogen, changes the fungal community composition, and reduces the fungal co-occurrence in the tomato rhizosphere.
what is the motivation for you work? improve the language
Answer: The motivation of this study was added here. Please see following (line95-97).
Our aims are to determine whether the fungal community composition, specifically the fungal pathogens, and the fungal networks were affected by R. solanacearum invasion in the rhizosphere of tomato plant.
- Material and methods
- this paragraph needs to be substantially improved the experimental design is not clear
Answer: Thanks for your suggestion. The parts of material and methods, specifically experimental design, were improved based on your following suggestion.
- Line 95-96: We randomly and widely took 72 tomato plants from 24 plots (total: 30; 12 plots for healthy plants and 12 plots for diseased plants)
this is not clear
Answer: I am sorry for the unclear information. The field was divided into 30 plots and we randomly and widely took 72 tomato plants from 24 plots (12 plots for healthy plants and 12 plots for diseased plants). The clear expression was added in the manuscript (line107-110).
- Line 96-98: We believe that the significant differences in the fungal community between healthy and bacterial wilt disease tomato plants were from R.solanacearum invasion
this is result how can you say this here?
Answer: We have deleted these sentences.
- Line 98-101: The detailed sampling method was listed as follows. At the end of the harvest (approximately 14 weeks for tomato plants), we selected twelve plots with healthy or bacterial wilt-diseased tomato plants at level-four disease severity [18] for sampling on October 11, 2016
This is also not clear
Answer: We have made it clear. Please see following
The detailed sampling method is listed as follows. At the end of the harvest (approximately 14 weeks for tomato plants, October 11, 2016), twelve plots with healthy or bacterial wilt-diseased tomato plants (level-four disease severity [19]) were selected to sample (line110-ll3).
- Line 101-106: Within each plot, rhizosphere soil samples from three tomato plants were collected and mixed into a single sample (Figure. 1). After pooling, 24 rhizosphere soil samples were obtained for soil DNA extraction.
reformulate
Answer: We have reformulated these sentences. Please see following(line113-117).
Within each plot, rhizosphere soil samples from three tomato plants were collected and mixed into a single sample (Figure 1). After the plants were dug out, the roots were carefully shaken to remove the loosely adhering soil, and the remaining attached soil was carefully collected as rhizosphere soil using a sterile brush. In total, 24 rhizosphere soil samples were obtained for soil DNA extraction.
- Line 107-108: To verify our main conclusion that R. solanacearum invasion increases the abundance of F. solani in the rhizosphere of bacterial wilt-diseased tomato plants,
this is discussion
Answer: we have deleted the sentences here.
- Line 109-113:We sampled healthy and bacterial wilt-diseased tomato plants in another soil, in which tomato and strawberry plants were planted under rotation conditions, and strawberry had been grown at the last planting cycle (October 2016 to March 2017), at the tomato florescence stage (April 26, 2017) in the same vegetable production center of Qilin Town.
reformulate
Answer: We have reformulated these sentences. Please see following (line118-122).
Another batch of healthy and bacterial wilt-diseased tomato plants were sampled in the soil, in which tomato and strawberry plants were planted under rotation conditions, and strawberry was grown at the previous planting cycle (October 2016 to March 2017). These tomato plants were sampled at the tomato florescence stage (April 26, 2017) in the same vegetable production center of Qilin Town.
- Line 119: To verify that the tomato wilt symptoms were caused by R. solanacearum, the pathogen R. solanacearum was isolated from diseased tomato stems
Start a new paragraph and improve the language
Answer: Yes, these sentences were started at a new paragraph, and the language was improved (line128-130).
To test that the tomato wilt symptoms were caused by R. solanacearum, the pathogen R. solanacearum was isolated from diseased tomato stems [20] on the selective media SMSA (Selective Medium South Africa).
- Line 120: Three R. solanacearum isolates were randomly selected from the SMSA selective medium and able to cause tomato bacterial wilt disease according to Koch’s Rule.
Tomato plants (number) were inoculated with
Answer: Three R. solanacearum isolates were randomly selected from the SMSA selective medium. Ten tomato seedlings (4-leaf stage) were inoculated with each R. solanacearum isolate, and all tested isolates can cause tomato bacterial wilt disease according to Koch’s Rule (line130-133).
- Line 123: DNA from the isolates was extracted and used as the template to amplify the 16S rRNA gene by PCR.
from the selective media or from plant samples?
Answer: Thanks for your reminding. DNA from the isolates was from the selective media (line134).
- Line 127-131: The BLAST results showed that the isolated strains from the diseased tomato stems shared 99.9% identity with the R. solanacearum GMI1000 strain (race 1, biovar 3). The phylogenetic tree of the isolates is shown in Figure. S10. The sequences were deposited at the NCBI GenBank (submission number: SUB5692774).
move this to Results part
Answer: We have moved this to results part. (See line 139)
- Line 137: After centrifugation at 8000 xg for 10 min,
do you mean rpm
Answer: No, it means rcf.
- Line 137: The supernatant was acidified to pH 2.5 with 5 M HCl to precipitate humic acid.
transferred to another flask and acidified
Answer: Yes, Thanks for your suggestions. We have added the information in the new manuscript (line144).
- Line 199: Soil culture experimental design
>1. why did you do this experiment?
Answer: We want to test whether the increased phenolic acid levels from diseased rhizosphere soils can facilitate the enrichment of Fusarium solani. Please see following.
To test whether the increased phenolic acid levels from the rhizosphere soil of bacterial wilt-diseased tomato can facilitate the enrichment of F. solani, we performed a soil culture experiment (line161-163).
>2. did you plant any tomato in it?
Answer: No, we did not plant tomato plants.
>3. did you do any artificial inoculation with the bacteria and the fungus?
Answer: No, we did not inoculate any bacteria and the fungus.
>4. Is this paragraph related to paragraph no. 2.2.? then combine them
Answer: Good idea! This paragraph was replaced after no. 2.2
- Line 216: Statistical analyses
This is very detailed paragraph try to reformulate and to shorten it
Answer: Thanks for your suggestions. We have reformulated and shortened this part.
- Results
- Line 258: Unless otherwise noted, “H” and “D” hereafter refer to the rhizosphere soils of healthy and bacterial wilt-diseased tomato plants, respectively
You mentioned that already see lines 201-202
Answer: We have moved the sentences.
- Line 260-262: R. solanacearum invasion significantly reduced the Shannon diversity of the fungal community (P=0.02). Principal coordinate analysis (PCoA) was performed based on Bray-Curtis index dissimilarity of relative abundance data. A clear difference was found between the fungal community structures of bacterial wilt-diseased and healthy tomato plant rhizospheres (Figure. 2b).
see 218-220
Answer: we have shortened and reformulated this part. Please see following (line284-288).
- solanacearum invasion significantly reduced the Shannon diversity of the fungal community (P=0.02) (Figure 2a). The results of the PCoA showed that a clear difference was found between the fungal community structures of bacterial wilt-diseased and healthy tomato plant rhizospheres (Figure 2b)
- Line 264: You did not say any thing regarding Fig. 2a, why?
Answer: Sorry for missing the legend of Fig. 2a. Fig. 2a presents the Shannon diversity. We have added it in the new manuscript (line285).
- Line 264-274: The indicator taxa of one treatment mean that such taxa had significant relationships with that treatment
This is not clear
Answer: The indicator taxa of one treatment mean that such taxa are characteristic in that treatment (line301).
- Line 278: Gibberella is the teleomorph of Fusarium and it is not used anymore in Fusarium taxonomy. The name Fusarium is used now to indicate to the anamorph and the teleomorph of Fusarium
Answer: Thanks for your suggestion. Gibberella is changed to Fusarium in the new manuscript.
- Line 283-286: The qPCR-determined copy numbers of the 16S RNA gene (total bacteria), internal transcribed spacer (ITS) region (total fungi), subunit of the flagellar filament gene (fliC) (R. solanacearum) and ITS I (F. solani) in the rhizosphere of bacterial wilt-diseased tomato plants were significantly higher than those in the rhizosphere of healthy tomato plants (Figure. 3).
see 187-189 you repeat the same story
Answer: We have shortened and reformulated these sentences (line308-311).
The qPCR-determined copy numbers of the 16S RNA gene (total bacteria), ITS (total fungi), fliC (R. solanacearum) and ITS I (F. solani) in the rhizosphere of bacterial wilt-diseased tomato plants were significantly higher than those in the rhizosphere of healthy tomato plants (Figure 3)
- Line 351: Three phenolic acids (p-hydroxybenzoic acid, vanillic acid, and ferulic acid) were determined in healthy and wilt-diseased tomato plant rhizospheres (Supporting information 1: Figure. S8).
see lines 147-151
Answer: We have detected six phenolic acids in the tested soils. Here, we want to show that only three phenolic acids are detected in the rhizospheres soils.
- Line 364: To determine which soil factor can significantly affect the abundance of dominant OTUs (top 20), we analyzed regression associations between soil factors and dominant OTUs by random forest analysis (Figure. 5).
see lines 239-241
Answer: we have removed this part.
- Discussion
- Line 401:Previous studies indicated that pathogenic bacterial and protist invasions affected soil microbial communities [4,17,30], and we found that the abundance of F. solani, a pathogenic fungus, increased in the rhizosphere soil of bacterial wilt-diseased tomato plants (Figure. 3).
this is repetition
Answer: we have deleted these sentences.
- Line401-409: Our findings support the following proposed model illustrated in Figure. 6a. R. solanacearum can cause brownish discoloration in the root vascular system, leading to varying degrees of root decay. An increased concentration of phenolic acids was found in the diseased tomato plant rhizosphere soils (Table 2). The soil culture experiment confirmed that such levels of phenolic acids can significantly stimulate a different fungal pathogen, F. solani growth (Figure. 6a).
this is results
Answer: It was used to explain the reasons of enrichment of F. solani. We have reformulated these sentences and combined them and pervious study to discuss phenolic acids can stimulate F. solani growth. Please see following (line429-435).
It is likely that R. solanacearum can cause brownish discoloration in the root vascular system, thereby leading to varying degrees of root decay and resulting in the release of phenolic acids. Indeed, we found that the three tested phenolic acids were increased in the rhizosphere of diseased tomato plant (Table 2) and the indigenous fungal pathogen F. solani was significantly enriched in soil amended with such levels of phenolic acids (Figure S9).
- Line411-414: In addition, it is known that R. solanacearum is strongly competitive, demonstrating phenotypic changes in response to different environments, high putrescine tolerance and enhanced pathogenicity under low iron conditions.
this is not clear and it is the first time you mention iron
Answer: Indeed, the iron information doesn't seem to fit our theme. We have reformulated these sentences and added the following sentence (line436-440).
In addition, the environment of rotten roots may be greatly different from that of heathy roots in the rhizosphere soil, such as by releasing the highly toxic substance putrescine. It is known that R. solanacearum is strongly competitive and demonstrates phenotypic changes in response to different environments as well as high putrescine tolerance.
- Line419-423: Moreover, the number of potential beneficial microbes that produce antagonistic substances against F. solani may also decrease. In our study, the copy number of the functional genes (sfr, fen, dfn and itu) related to the synthesis of antagonistic substances against F. solani, indeed decreased in the wilt-diseased rhizosphere soils and had negative associations with the copy number of ITS I (F. solani) suggesting that the decreased abundances of antagonistic substances produced by such genes stimulated the growth of F. solani.
is it your results or the results of the reference?
taking the whole sentence into account, this is too long and difficult to understand
Answer: the copy number of functional genes (sfr, fen, dfn and itu) is the result in our manuscript. We have re reformulated these sentences. Please see followings (line444-449)
Moreover, the copy number of the functional genes (sfr, fen, dfn and itu) related to the synthesis of antagonistic substances against F. solani [30] was significantly decreased in the wilt-diseased rhizosphere soils (Figure 3) and had negative associations with the copy number of ITS I (F. solani) (Figure 6a) suggesting that the decreased abundance of antagonistic substances produced by such genes stimulated the growth of F. solani.
- Line 424: To test the robustness of our results, we sampled bacterial-wilt diseased and healthy samples in another soil.
you already mentioned this different times
Answer: We have checked the sentence is only shown here.
- Line 425-428: The results confirmed that R. solanacearum invasion can increase the abundance of F. solani, regardless of the history of soil use and tomato plant growth stage, generalizing our results (Supporting information 1: Figure. S3).
this is results
Answer: Thanks for your suggestion. The sentence is an extension of the result and important to support our conclusion.
- Line 429: such as Plectosphaerella,
this is not enough to say that they are tomato pathogens, you have to define the species
Answer: The species names were added in th sentence, such as Plectosphaerella cucumerina and Verticillium dahlia (line456).
- Line 432-437: It is likely that a consortium of fungal pathogens was enriched in the rotten rhizosphere environment caused by R. solanacearum invasion. It is known that rotten plant tissue can stimulate soil pathogen growth. The result indicated an increased the potential risk of host infection in the wilt-diseased soil. These potential pathogens are needed to quantify by qPCR in a future study.
reformulate
Answer: We have reformulated these sentences. Please see following (line459-464).
It is known that rotten plant tissue can stimulate soil pathogen growth. Thus, a consortium of fungal pathogens was likely enriched in the rotten rhizosphere environment caused by R. solanacearum invasion. This result indicated an increased the potential risk of host infection by fungal pathogen in the wilt-diseased soil. These potential pathogens must be quantified by qPCR in a future study.
- Line 441-444: It is likely that the rhizosphere environment was changed by R. solanacearum invasion, which might have led to root decay and consequently the release of nutrients into the soil. Indeed, the increased contents of total C, N and phenolic acids were observed in the wilt-diseased rhizosphere soils (Table 2), and these factors had significant associations with fungal communities (Supporting information 1: Table. S3).
this is repetition
Answer: We have reformulated these sentences and deleted the repetitive parts. Please see following (line466-468).
It is likely that the rhizosphere environment was changed by R. solanacearum invasion, which might have led to root decay and consequently the release of nutrients into the soil as observed in Table 2.
- Line 449-455: Moreover, the fungal pathogen F. solani was positively affected by the increased content of p-hydroxybenzoic acid (Figure. 5). In addition, the enrichment of potential pathogens had the significant associations with the changes in the fungal community structure (Figure. 6b). It is possible that the enriched fungi may cause increased niche occupation in the wilt-diseased rhizosphere soil, decreasing the abundances of other fungi.
this is results
Answer: We have deleted the results and reformulated these sentences (lines471-474).
Thus, the changes in nutrients, such as phenolic acids, may facilitate the enrichment of potential fungal pathogens, thereby resulting in the changes in fungal communities as observed by the associations in our model (Figure 6b).
- Line 455-460: Although these significant associations between enriched certain microbes and changes in the fungal community structure from our model were obtain by statistical approach (Figure. 6), the fake associations may still exist and this problem cannot be solved in the current statistical approach. However, our main conclusion that R.solanacearum invasion increases the phenolic acid contents resulting in favoring colonization by the pathogenic fungi F. solani was not affected by such problem.
reformulate
Answer: We have reformulated these sentences. Please see following (line474-479).
Although fake associations between enriched certain microbes and changes in the fungal community structure may still exist in our model (Figure 6b) and this problem cannot be solved via the current statistical approach, our main conclusion that R. solanacearum invasion increases the phenolic acid contents, thereby favoring colonization by the pathogenic fungi F. solani, was not affected by the observed problem.
- Line 461: In this study, the highest soil pH was found in the rhizosphere soil of wilt-diseased tomato plants (Table 2).
results
Answer: Thanks for your suggestion. We have to show the important result. We think it is the beginning of explaining the reasons of the result.
- Line 462-463: It is likely that the content of ammonia might have increased during the root decay process, leading to an increase in the soil pH.
you can not say that without reference
Answer: Thanks for your reminding. The following reference was added in manuscript.
Ingelög, T.; Nohrstedt, H.Ö. Ammonia formation and soil pH increase caused by decomposing fruitbodies of macrofungi. Oecologia 1993, 93, 449-451, doi:10.1007/BF00317891.
- Line 468: However, no significant differences in soil pH were found between the healthy and wilt-diseased tomato plant rhizospheres in the study of Wei et al.
what about the pH at the last sampling date?
Answer: about 5.1
- Line 472-475: In addition, high pH and total C content reportedly increased fungal diversity. However, low fungal diversity was found in the wilt-diseased rhizosphere soils.
reformulate
Answer: We have reformulated these sentences. Please see following (line489-492).
In addition, the high pH and total C content reportedly increased fungal diversity. However, low fungal diversity was found in the wilt-diseased rhizosphere soils with high pH and total C content.
- Line 480-486: In agreement with our results, studies have shown that R. solanacearum invasion reduces the bacterial network complexity of tomato [4] and tobacco [17] plant rhizosphere soils. It is known that complex co-occurrences have more links than simple co-occurrences; that is, the former is more robust against environmental factors than the latter [40]. Hub taxa (highly connected nodes) may play important roles in maintaining microbial community stability and coordinating many relationships throughout the microbiome.
improve the language
Answer: We have improved the language. Please see the following (line500-506).
In agreement with our results, studies have shown that R. solanacearum invasion reduces the bacterial network complexity of tomato and tobacco plant rhizosphere soils. It is known that complex co-occurrences are more robust against environmental factors than simple co-occurrences. Moreover, hub taxa (highly connected nodes) may play important roles in maintaining microbial community stability and coordinating many relationships throughout the microbiome
- Line 488-490: Mortierella can increase the resistance of plants to phytopathogens by the production of polyunsaturated fatty acids
reformulate
Answer: We have reformulated these sentences. Please see following (line508-509).
Mortierella hygrophila can induce vine plant defense responses to powdery mildew disease by the production of polyunsaturated fatty acids
- Line 491-497: This combined information suggests that certain potentially beneficial fungi, such as Mortierella, increase the co-occurrence complexity and form cooperative associations with other taxa to stimulate plant host growth. For example, the combined application of arbuscular mycorrhizal fungi and Mortierella yielded a better effect than the application of arbuscular mycorrhizal fungi alone on the increase in the shoot and root growth of host plants under salt stress
reformulate
Answer: We have reformulated these sentences. Please see following (line511-516).
These results suggest that the potentially beneficial fungi Mortierella may form cooperative associations with other taxa to stimulate plant host growth. For example, the combined application of arbuscular mycorrhizal fungi and Mortierella yielded a better effect on the increase in the shoot and root growth of Kosteletzkya virginica plants than the application of arbuscular mycorrhizal fungi alone
- 19. Line 502-504: Because the inference co-occurrence was built by the relative abundances of OTUs, certain biases may exist.
reformulate
Answer: We have reformulated the sentences. Please see following (line525).
It is likely that the inference co-occurrence was built by the relative abundances of OTUs resulting in certain biases.
- Line 505: Ralstonia solanacearum invasion reduced the fungal co-occurrence complexity (Figure. 4).
results
Answer: Thanks for your suggestion. Again, we have to show the important result. We think it is the beginning of explaining the reasons of the important result.
- Line 506-509: It is possible that the enriched microbes, such as R. solanacearum and F. solani, occupy more niches, leading to the disappearance of certain members of the fungal community, as shown by the negative association between the fungal diversity and the copy numbers of fliC (R. solanacearum) and ITS I (F. solani).
repetition and too long
Answer: Here we highlighted the effects of enrichment of R. solanacearum and F. solani on fungal diversity, and we have reformulated the sentences (line527-530).
It is possible that the enriched microbes, such as R. solanacearum and F. solani, occupy more niches, thereby leading to the disappearance of certain members of the fungal community. A negative association between the fungal diversity and copy numbers of fliC (R. solanacearum) and ITS I (F. solani) was observed in Figure 6a.
- Line 513-514: Moreover, it was known that a low diversity microbial community resulted in a simple microbial co-occurrence
???
Answer: In general, if the microbial community with lower diversity, the links among the microbes were less.
- Line 515: Overall, the enriched microbes stimulated by the nutrients, such as phenolic acids, from the rotten environment occupied more niches, resulted in a low fungal diversity and simple fungal co-occurrence
are they nutrient?
Answer: Yes, they are. Certain fungal pathogens, such as, Fusarium can use phenolic acids as nutrients.
- Line 518: In line with our study, when maize residues were applied to soil, the relative abundance of Fusarium graminearum increased, and the complexity of bacteria-fungi networks decreased
you can not compare with F. graminearum on maize it is different pathosystem on maize residues
Answer: The sentences were deleted.
- Line 521-524: It is likely that other substances, such as cellulose, from rotten root environments and the decreased abundance of antagonistic genes also stimulated the growth of Fusarium, and the multiple stimulation effects may resulted in a great Fusarium abundance to reduce the fungal diversity.
this is not clear
Answer: We have made it clear. Please see following (line536-539).
It is likely that other substances from rotten root environments, such as cellulose, and the decreased abundance of antagonistic genes resulted in great enrichment of Fusarium which occupied niches to reduce the fungal diversity.
- Line 524-527: When the fungal networks were constructed with shared OTUs, the fungal network complexity of the diseased rhizosphere community was also lower than that of the healthy rhizosphere fungal network community (Supporting information 1: Figure. S7).
results
Answer: Yes, thank for your reminding. We have to show certain results and combine certain reasons and previous studies to discuss the theme.
- Line 527-531: Due to root decay caused by the wilt disease pathogen, the nutrient content in wilt-diseased tomato plant rhizosphere soils was higher than that in healthy tomato plant rhizosphere soils, as shown by the increase in the total C and N and phenolic acid contents in wilt-diseased tomato plant rhizosphere soils. Under resource-limited conditions, cooperative associations among microbes are important for survival, and microbes form more connecting links, resulting in a complex network. In line with the results of our study, a complex fungal network was found in the rhizosphere soil of an organic farm. The application of organic amendments containing lower immediate nutrient resources resulted in more links among the fungal communities.
too long and add reference
Answer: We have shortened these sentences and added reference. Please see following (line542-547).
It is likely that lower nutrient contents, such as total C and N, and phenolic acid contents (Table 2) resulted in the enhancement of cooperative associations among microbes under resource-limited conditions in heathy tomato plant rhizosphere soils. In line with the results of our study, the application of organic amendments containing lower immediate nutrient resources resulted in more links among the fungal communities in bulk soils
Conclusions
- Line 545-549: In addition, we analysed the associations between changes in fungal communities and R. solanacearum invasion and soil factors and found that the cascade changes in rhizosphere compounds and abundances of potential fungal pathogens triggered by R. solanacearum invasion may be the main reasons for the changes in fungal communities
too long
Answer: We have shorten these sentences. Please see the following (line556-559).
In addition, the cascade changes in rhizosphere compounds and abundances of potential fungal pathogens triggered by R. solanacearum invasion may be the main reasons for the changes in fungal communities.
- Line 551-555: Overall, we conclude that the invasion of tomato plant roots by R. solanacearum increases phenolic acid contents, decreases antagonistic gene copy numbers and fungal diversity in the rhizosphere soil, which in turn may result in increased colonization by the pathogenic fungi F. solani, and simple fungal co-occurrence.
repetition
Answer: These sentences were deleted.
Reviewer 2 Report
This research was focused on the effect of bacterial pathogen on the compounds in rhizosphere and fungal pathogen.
Combing analysis with sequence data and compound data showed the whole rhizosphere community.
some changes of each factor has been already reported, but the study about their relationship was poor, you provided new insight.
However, some description should be improved.
Random forest analysis showed the interesting relationship between OTUs and soil factors.
You revealed the correlation between F. solani and ferulic acid and vanillic acid.
But this description is inadequate to determine whether this analysis is appropriate.
There is no information what is top 20 OTUs.
It is not shown how difference two Ascobolus are in Figure 5 (they have different IDs?).
I couldn't understand it.
This analysis is very important point in this manuscript, so you should describe more politely.
In figure 6, you describe the relationship as arrow, which indicates the direction of effect.
Although your research unrevealed the correlation, it is unclear which of two factor is the cause.
You had better that arrow change to just bar or double-arrow.
minor point:
No description of figure. 2a in main text.(maybe line.260)
Author Response
Answer to the reviewers:
Dear Reviewers,
Thanks for reviewing our manuscript during the epidemic period.
Major modifications were listed as follows.
- We checked the annotation information of OTUs and found large tomato ITS sequences (up to approximately 73.47%) existed in certain samples. Amplification of so many non target sequences may consume a lot of PCR materials leading to the low sequences for target sequences that may not meet the fungal community analysis condition. Thus, we deleted these samples (3 samples for healthy plants and 3 samples for diseased plants) and redo the fungal community analysis with only fungal sequences.
- Because the website of network tool Ecological Network Analyses Pipeline (MENA) cannot be opened, we used Sparcc to perform the network analysis.
Although these changes are in the new version, our main conclusion is the same.
- The new manuscript was checked by professional English editing service company (AJE)
Thanks again for your suggestions!
This research was focused on the effect of bacterial pathogen on the compounds in rhizosphere and fungal pathogen. Combing analysis with sequence data and compound data showed the whole rhizosphere community. Some changes of each factor have been already reported, but the study about their relationship was poor, you provided new insight. However, some description should be improved.
- Random forest analysis showed the interesting relationship between OTUs and soil factors. You revealed the correlation between F. solani and ferulic acid and vanillic acid. But this description is inadequate to determine whether this analysis is appropriate.
Answer: Thanks for your suggestions. In the new version of the manuscript, we used the increased mean square error (%IncMSE) to evaluate the importance of the soil factors and only selected the significant soil factors (p value of %IncMSE<0.05) for further analysis. To present the results of the regression comprehensively, the IncMSE value and R2 were added (Table S5 in Supporting information 1). The higher IncMSE value, the more important of the soil factor. R2 (R-squared) is used to explain how many variation can be explained by the regression model. The higherR2 value, the more precise result of regression equation.
- There is no information what are top 20 OTUs ?
Answer: The top 20 OTUs have been shown in Table.S4 in Supporting information 1.
- It is not shown how difference two Ascobolus are in Figure 5 (they have different IDs?). I couldn't understand it.
Answer: The two Ascobolus mean they have different OTU identity, what now is added before the genera names (line774, page32).
- This analysis is very important point in this manuscript, so you should describe more politely.
Answer: We added the background of the Random forest and the meanings of the %IncMSE in the method section, lines251-262, page11.
We performed a random forest analysis to determine the regression associations between soil factors and dominant OTUs (top 20). Random forest analysis can provide high prediction accuracy by building the decision trees based on bootstrapped samples. This analysis is a nonlinear statistical and nonparametric method without prior distributional assumptions. Certain samples used to train the model are called in-bag data, and the other samples are termed out-of-bag data. Trees fully grown are used to predict the out-of-bag data, and the importance of the variable is obtained by randomly permuting the values of that variable for the out-of-bag data and calculating increases in the mean squared error (%IncMSE). The higher the IncMSE value, the more important the variable. In our study, a random forest analysis was performed with 999 permutations using the randomforest and rfPermute packages and visualized using Gephi
- In figure 6, you describe the relationship as arrow, which indicates the direction of effect. Although your research unrevealed the correlation, it is unclear which of two factor is the cause. You had better that arrow change to just bar or double-arrow.
Answer: We agree with the reviewer. Indeed, our result cannot show the direction of effect. We have changed the arrow to bar in Figure. 6.
- Minor point:
No description of figure. 2a in main text.(maybe line.260)
Answer: We had added the description of Figure. 2a in main text, lines285, page12.
Figure. 2a: Fungal community Shannon diversity of rhizosphere soils between healthy (H) and bacterial wilt-diseased (D) tomato plant rhizospheres. Statistical significance was determined based on Student’s t test. * P < 0.05.”
Reviewer 3 Report
In presented article, authors tried to find relation between bacterial tomato wilt and Fusarium solani infection of tomato. They analyzed difference between rizosphere community of healtly and infected plants, occurence of Ralstonia and fusarium specific markers, markers for antagonism and occurence of phenolic acids. Differences were confirmed and role of phenolic acids was discussed.
Article is hypothesis driven. Despite hypothesis about increased occurence of oportunistic fungi F. solani in plants weakened by Ralstonia is not very impressive, analysis of phenolic acid role in pathogenesis, although previously determined, seems to be inovative for me. Absolutely, study is complex and should be published.
Methods
I wonder how authors found fields with diseased plants. Were they planted intentionally?
I don’t understand sentence "we randomly and widely took 72 tomato plants from 24 plots (total: 30; 12 plots for healthy plants and 12 plots for diseased plants)." what is total: 30?
Primers ITS1R was developed by white et al in 1993 and it should be called ITS2. ITS1 region is spacer between 18S and 5.8S rRNA, not 18S rRNA spacer. ITS1 sequences are very variable can be as short as 8bp and excessive triming is not recommended for them.
I am not very familiar with mothur, but what is quality score 0.5? I did not found it in mothur manual. Phred Q30 quality is considered to be standard for filtering. Regardless of this article I recommend authors to use more modern pipeline, especially DADA2 works fine.
Why authors used Chao1 index while removing singletons. As this index is based on singletons and doubletons, it suffers from singleton removing. On the other side removing of low copy sequences is the right way to use Illumina results.
regarding fig.2c Many of sequences are not recognized. Try to exclude sequences of tomato ITS. Use of other then fungal database may solve this problem too. Sometimes many non-fungal sequences fall into ascomycota when only fungal database is used.
figure3 I wonder that copy numbers of genes had normal distribution required for Student`s test. Was normality tested? I found "yes". OK, no more questions.
Figure 6 suggest causality between phenolic acid and R. solanacearum or F. solani. Although proposed model looks logical this is not proved. Arrows should be used very carefully according my opinion and in the right side of figure some arrows should be certainly deleted. For better understanding, a study monitoring changes of fungal community or phenolic acid contents alongside to Ralstonia development should help in proposing/confirming such model.
In the discussion authors frequently repeated parts from results. On the other side results seems to be too long in some parts and will benefit from shortening. Authors also used complicated sentences, sometimes confusing.
Author Response
Answer to the reviewers:
Dear Reviewers,
Thanks for reviewing our manuscript during the epidemic period.
Major modifications were listed as follows.
- We checked the annotation information of OTUs and found large tomato ITS sequences (up to approximately 73.47%) existed in certain samples. Amplification of so many non target sequences may consume a lot of PCR materials leading to the low sequences for target sequences that may not meet the fungal community analysis condition. Thus, we deleted these samples (3 samples for healthy plants and 3 samples for diseased plants) and redo the fungal community analysis with only fungal sequences.
- Because the website of network tool Ecological Network Analyses Pipeline (MENA) cannot be opened, we used Sparcc to perform the network analysis.
Although these changes are in the new version, our main conclusion is the same.
- The new manuscript was checked by professional English editing service company (AJE)
Thanks again for your suggestions!
In presented article, authors tried to find relation between bacterial tomato wilt and Fusarium solani infection of tomato. They analyzed difference between rhizosphere community of healthy and infected plants, occurence of Ralstonia and fusarium specific markers, markers for antagonism and occurence of phenolic acids. Differences were confirmed and role of phenolic acids was discussed.
Article is hypothesis driven. Despite hypothesis about increased occurence of oportunistic fungi F. solani in plants weakened by Ralstonia is not very impressive, analysis of phenolic acid role in pathogenesis, although previously determined, seems to be innovative for me. Absolutely, study is complex and should be published.
Answer: Thank you for the positive comments.
Methods
- I wonder how authors found fields with diseased plants. Were they planted intentionally?
Answer: The fields belong to a vegetable production center that mainly planted tomato and had cooperation with our university. Because the field is under tomato continuous cropping condition, tomato bacterial wilt disease occurred naturally.
- I don’t understand sentence "we randomly and widely took 72 tomato plants from 24 plots (total: 30; 12 plots for healthy plants and 12 plots for diseased plants)." what is total: 30?
Answer: We are sorry for the unclarity. Total 30 means that the field was divided to 30 plots. We have made it clear in the manuscript (line 107).
The experimental field was divided into 30 plots. To weaken the previous differences in the fungal community of field soils, we randomly and widely took 72 tomato plants from 24 plots (12 plots for healthy plants and 12 plots for diseased plants).
- Primers ITS1R was developed by white et al in 1993 and it should be called ITS2. ITS1 region is spacer between 18S and 5.8S rRNA, not 18S rRNA spacer. ITS1 sequences are very variable can be as short as 8bp and excessive triming is not recommended for them.
Answer: Thanks for the information. We have checked the primers sequences in the literature. Indeed, ITS1R should be called ITS2. The correct primer name is added in the manuscript (line191-193).
ITS1 region is spacer between 18S and 5.8S rRNA, not 18S rRNA spacer
Answer: correcte. Line 194
ITS1 sequences are very variable can be as short as 8bp and excessive triming is not recommended for them.
Answer: The length of ITS1 is about 350bp. Indeed, the ITS1 sequences can be very variable, therefore, we considered only sequences longer than 200bp in our analysis.
- I am not very familiar with mothur, but what is quality score 0.5? I did not found it in mothur manual. Phred Q30 quality is considered to be standard for filtering. Regardless of this article I recommend authors to use more modern pipeline, especially DADA2 works fine.
Answer: Quality score 0.5 was from USEARCH, not Mothur. It means that if the expected errors (in short, sum of incorrected base call) of reads were more than 0.5, the reads will be deleted. If you want to know the detailed information about expected errors, please see the website: http://www.drive5.com/usearch/manual/exp_errs.html.
It is very well appreciated the reviewer recommendation to use DADA2 pipeline for our future amplicon sequence analyses. Thank you.
- Why authors used Chao1 index while removing singletons. As this index is based on singletons and doubletons, it suffers from singleton removing. On the other side removing of low copy sequences is the right way to use Illumina results.
Answer: After we looked over the formula of Chao1, indeed, Chao1 index is based on singletons and doubletons. Thus, we only used Shannon index in the new manuscript.
- regarding fig.2c Many of sequences are not recognized. Try to exclude sequences of tomato ITS. Use of other then fungal database may solve this problem too. Sometimes many non-fungal sequences fall into ascomycota when only fungal database is used.
Answer: Thanks for your suggestion. We checked the annotation information of OTUs and found large tomato ITS sequences (up to approximately 73.47%) existed in certain samples. Thus, we deleted these samples and combined UNITE Fungal ITS and Warcup Fungal ITS database to classify the OTUs and rerun the analysis.
- figure3 I wonder that copy numbers of genes had normal distribution required for Student`s test. Was normality tested? I found "yes". OK, no more questions.
Figure 6 suggest causality between phenolic acid and R. solanacearum or F. solani. Although proposed model looks logical this is not proved. Arrows should be used very carefully according my opinion and in the right side of figure some arrows should be certainly deleted.
Answer: Indeed, the associations in Figure 6 was only based on the statistical approach, the direction of effect is hard to tell. Thus, we replace the arrows with bars.
- For better understanding, a study monitoring changes of fungal community or phenolic acid contents alongside to Ralstonia development should help in proposing/confirming such model.
Answer: Thanks for your suggestions. The associations between abundances of Ralstonia and phenolic acid contents or fungal community were tested by Mantel tests. The R value is 0.9 and 0.8, respectively and shown in the Fig.6.
- In the discussion authors frequently repeated parts from results. On the other side results seems to be too long in some parts and will benefit from shortening. Authors also used complicated sentences, sometimes confusing.
Answer: Thanks for your suggestions. We have removed the repetitive results in discussion and the new manuscript was checked by professional English editing service company (AJE).
Round 2
Reviewer 1 Report
See my comments in the pdf file
